# Unsupervised Metric Learning via Nonlinear Feature Space Transformations

## Abstract

In this paper, we propose a nonlinear unsupervised metric learning framework to boost of the performance of clustering algorithms. Under our framework, nonlinear distance metric learning and manifold embedding are integrated and conducted simultaneously to increase the natural separations among data samples. The metric learning component is implemented through feature space transformations, regulated by a nonlinear deformable model called Coherent Point Drifting (CPD). Driven by CPD, data points can get to a higher level of linear separability, which is subsequently picked up by the manifold embedding component to generate well-separable sample projections for clustering. Experimental results on synthetic and benchmark datasets show the effectiveness of our proposed approach over the state-of-the-art solutions in unsupervised metric learning.

## 1 Introduction

Cluster analysis has broad applications in various disciplines. Grouping data samples into categories with similar features is an efficient way to summarize the data for further processing. In measuring the similarities among data samples, the Euclidean distance is the most common choice in clustering algorithms. Under Euclidean distance, feature components are assigned with the same weight, which essentially assumes all features are equally important across the entire data space. In practice, such setup is often not optimal. Learning a customized metric function from the data samples can usually boost the performance of various machine learning algorithms (Bellet et al., 2013). While metric learning has been extensively researched under supervised (Xing et al., 2003; Weinberger & Saul, 2009; Wang et al., 2012; Noh et al., 2010) and semi-supervised settings (Peng et al., 2002; Domeniconi et al., 2001; Zhang et al., 2016; Niu et al., 2014), unsupervised metric learning (UML) remains a challenge, in part due to the absence of ground-truth label information to define a learning optimality. In this paper, we focus on the problem of UML for clustering.

As the goal of clustering is to capture the natural separations among data samples, one common practice in the existing UML solutions is to increase the data separability and make the separations more identifiable for the ensuing clustering algorithm. Such separability gain can be achieved by projecting data samples onto a carefully chosen low-dimensional manifold, where geometric relationships, such as the pairwise distances, are preserved. The projections can be carried out linearly, as through the Principle Component Analysis, or nonlinearly, as via manifold learning solutions. Under the dimension-reduced space, clustering algorithms, such as $K$-means, can then be applied.

Recent years have seen the developments of UML solutions exploring different setups for the low-dimensional manifolds. FME (Nie et al., 2010) relies on the learning of an optimum linear regression function to specify the target low-dimensional space. Abou-Moustafa et al. (2013) model local sample densities of the data to estimate a new metric space, and use the learned metric as the basis to construct graphs for manifold learning. Application-specific manifolds, such as Grassmann space (Huang et al., 2015) and Wasserstein geometry (Seguy & Cuturi, 2015), have also been studied. When utilized as a separate preprocessing step, dimensionality reduction UML solutions are commonly designed without considering the ensuing clustering algorithm and therefore cannot be fine-tuned accordingly.

AML (Ye et al., 2007) takes a different approach, performing clustering and distance metric learning simultaneously. The joint learning under AML is formulated as a trace maximization problem, and numerically solved through an EM-like iterative procedure, where each iteration consists of a data

projection step, followed by a clustering step via kernel $K$-means. The projection is parameterized by an orthogonal, dimension-reducing matrix. A kernelized extension of AML was proposed in (Chen et al., 2007). As the projection models are built on linear transformations, their capabilities to deal with complex nonlinear structures are limited.

UML solutions performing under the original input space have also been proposed. SSO (Jiang et al., 2011) learns a global similarity metric through a diffusion procedure that propagates smooth metrics through the data space. CPCM (Gupta et al., 2008) relies on the ratio of within cluster variance over the total data variance to obtain a linear transformation, aiming to improved data separability. As the original spaces are usually high-dimensional, UML solutions in this category tend to suffer from the local minima problem.

In light of the aforementioned limitations and drawbacks, we propose a new nonlinear UML framework in this paper. Our solution integrates nonlinear feature transformation and manifold embedding together to improve the data separability for $K$-means clustering. Our model can be regarded as a fully nonlinear generalization of AML, in which the transformation model is upgraded to a geometric model called *Coherent Point Drifting* (CPD) (Myronenko & Song, 2010a). Data points are driven by CPD to reach a higher level of linear separability, which will be subsequently picked up by the manifold embedding component to generate well-separable sample projections. At the end, $K$-means is applied on the transformed, dimension-reduced embeddings to produce label predictions. The choice of CPD is with the consideration of its capability of generating high-order yet smooth transformations. The main contributions of this paper include the following.

- Our proposed fully nonlinear UML solution enhances data separability through the combination of CPD-driven deformation and spectral embeddings.
- To the best of our knowledge, this is the first work that utilizes dense, spatial varying deformations in unsupervised metric learning.
- The CPD optimization has a closed-form solution, therefore can be efficiently computed.
- Our model outperforms state-of-the-art UML methods on six benchmark databases, indicating promising performance in many real-world applications.

The rest of this paper is organized as follows. Section 2 describes our proposed method in detail. It includes the description of CPD model, formulation of our CPD based UML, optimization strategy and the approach to kernelize our model. Experimental results are presented in Section 3 to validate our solutions with both synthetic and real-world datasets. Section 4 concludes this paper.

## 2 Unsupervised Metric Learning through CPD Transformations (CPD-UML)

Many machine learning algorithms have certain assumption regarding the distribution of the data to be processed. $K$-means always produces clustering boundaries of hyperplanes, working best for the data set made of linearly separable groups. For data sets that are not linearly separable, even they are otherwise well-separable, $K$-means will fail to deliver. Nonlinearly displacing the data samples to make them linearly separable would provide a remedy, and learning such a transformation is the goal of our design. The application of such a smooth nonlinear transformation throughout feature space (either input space or kernel space) would change pairwise distances among samples, which is equivalent to assigning spatially varying metrics in different areas of the data space.

In our framework, the CPD model is chosen to perform the transformation. Originally designed for regulating points matching, CPD moves the points $\mathcal{U}$ towards the target $\mathcal{V}$ by estimating an optimal continuous velocity function $v(x) : \mathbb{R}^d \to \mathbb{R}^d$ under Tikhonov regularization framework: $\mathcal{T}[v] = \frac{1}{2} \sum_{i=1}^{n} [v_i - (u_i + v(u_i))]^2 + \frac{1}{2}\lambda||Lv||^2$, where $n$ is the number of samples in the dataset, and $d$ is the data dimension. $L$ represents a linear differentiation operator, and $\lambda$ is the regularization parameter. The regularization term in CPD is a Gaussian low-pass filter. The optimal solution $v(x)$ to matching $\mathcal{U}$ and $\mathcal{V}$ can be written in the matrix format as (Myronenko & Song, 2010b):

$$v(x_i) = \Psi \begin{pmatrix} g(x_i, x_1) \\ \cdots \\ g(x_i, x_n) \end{pmatrix} = \Psi \mathcal{G}(x_i, X), \quad (1)$$

where $\Psi$ (size $d \times n$) is the weight matrix for the Gaussian kernel functions, $g(x_i, x_j) = e^{-\frac{(x_i - x_j)^2}{2\sigma^2}}$. $\sigma$ is the width of the Gaussian filter, which controls the smoothness level of the deformation field.

## 2.1 Formulation of CPD-UML

Let $X = \{x_i | \ x_i \in \mathbb{R}^d, i = 1, \cdots, n\}$ denote a dataset. $K$-means clustering aims to partition the samples into $K$ groups $S = \{S_1, S_2, ..., S_K\}$, through the minimization of the following objective function:

$$\min_S \quad J = \sum_{c=1}^{K} \sum_{x_i \in S_c} ||x_i - \mu_c||^2 \qquad \text{where } \mu_c = \sum_{x_i \in S_c} x_i / n_c \tag{2}$$

$S_c$ is the set of data samples in the $c$-th cluster. $n_c$ is the number of data instances in cluster $S_c$, and $\mu_c$ is the mean of $S_c$.

Allowing samples to be moved, we intend to learn a spatial transformation to improve the performance of $K$-means clustering by making groups more linearly separable, as well as by harnessing the updated distance measure under the transformed feature space. Let $x_i$ be the initial location of an instance. Through the motion in Eqn. (1), $x_i$ will be moved to a new position $x_i^1$:

$$x_i^1 = x_i + v(x_i) = x_i + \Psi \mathcal{G}(x_i, X) \tag{3}$$

With Eqn. (3), Eqn. (2) can be reformulated as:

$$\min_{S^1, \Psi} \quad J = \sum_{c=1}^{K} \sum_{x \in S_c^1} ||x^1 - \mu_c^1||^2 \qquad \text{where } \mu_c^1 = \sum_{x^1 \in S_c^1} x^1 / n_c; \quad x^1 = x + \Psi \mathcal{G}(x, X) \tag{4}$$

Now $S^1 = \{S_1^1, S_2^1, ..., S_K^1\}$ is a partition of the transformed dataset. $\mu_c^1$ is the mean vector of the instances in cluster $S_c^1$. Our proposed CPD based unsupervised metric learning (CPD-UML) is designed to learn a spatial transformation $\Psi$ and a clustering $S^1$ at the same time. Eqn. (4) can be reformulated into a matrix format through the following steps. First, put the input dataset into a $d$-by-$n$ data matrix. Second, define a Gaussian kernel function matrix for the CPD deformation as:

$$G = G(X, X) = \{\mathcal{G}(x_1, X), \mathcal{G}(x_2, X), ..., \mathcal{G}(x_n, X)\} \tag{5}$$

The size of $G$ is $n$-by-$n$. Third, let $p$ be a vector of dimension $n_c$-by-$1$ with all elements equal to one, then the mean of the data instances within a cluster $S_c^1$ can be written as $\mu_c^1 = S_c^1 p / n_c$ (Zha et al., 2001). With these three formulations, and let $E$ be a permutation matrix, Eqn. (4) can be rewritten as:

$$\min_{S^1, \Psi} \quad J = \sum_{c=1}^{K} ||S_c^1 - \mu_c^1 p^T||_F^2 = \sum_{c=1}^{K} ||S_c^1 - S_c^1 pp^T / n_c||_F^2$$
$$S^1 = EX^1 = E(X + \Psi G(X, X)) \tag{6}$$

where $X^1$ is the transformed data matrix. Since $||A||_F^2 = \text{trace}(A^T A)$, Eqn. (6) can be written in the form of the trace operation:

$$\min_{S^1, \Psi} \quad J = \sum_{c=1}^{K} \text{trace}((S_c^1(I - pp^T / n_c))^T (S_c^1(I - pp^T / n_c)))$$
$$= \sum_{c=1}^{K} \text{trace}(S_c^1(I - pp^T / n_c)(I - pp^T / n_c)^T (S_c^1)^T) \tag{7}$$

As $\text{trace}(AB) = \text{trace}(BA)$, and $p^T p = n_c$, the $J$ in Eqn. (7) can be further reformulated as:

$$J = \sum_{c=1}^{K} \text{trace}((I - pp^T / n_c)(I - pp^T / n_c)^T (S_c^1)^T S_c^1)$$
$$= \sum_{c=1}^{K} \text{trace}((S_c^1)^T S_c^1 - (p^T / \sqrt{n_c})(S_c^1)^T S_c^1(p / \sqrt{n_c})) \tag{8}$$

Similar to (Zha et al., 2001), we define a $n$-by-$k$ orthonormal matrix $Y$ as the cluster indicator matrix:

$$Y = [Y_1, Y_2, ..., Y_K] \qquad \text{where } Y_c = (0, ...0, p, 0...0)^T / \sqrt{n_c} \tag{9}$$

With $X^1 = X + \Psi G(X, X)$ and the cluster indicator matrix in Eqn. (9), Eqn. (8) can be written into the following:

$$\min_{Y, \Psi} \quad J = \text{trace}((X + \Psi G)^T (X + \Psi G)) - \text{trace}(Y^T (X + \Psi G)^T (X + \Psi G)Y) \tag{10}$$

To reduce overfitting, we add the squared Frobenius norm $\lambda ||\Psi||_F^2 = \lambda \text{trace}(\Psi^T \Psi)$, to penalize any non-smoothness in the estimated transformations. $\lambda$ is a regularization parameter. Finally, our nonlinear CPD-UML solution is formulated as a trace minimization problem, parameterized by $Y$ and $\Psi$:

$$\min_{Y, \Psi} \quad J = \text{trace}((X + \Psi G)^T (X + \Psi G)) - \text{trace}(Y^T (X + \Psi G)^T (X + \Psi G)Y) + \lambda \text{trace}(\Psi^T \Psi) \tag{11}$$

## 2.2 OPTIMIZATION STRATEGY

To search for the optimal solutions of $Y$ and $\Psi$, an EM-like iterative minimization framework is adopted to update $Y$ and $\Psi$ alternatingly. The transformation matrix $\Psi$ is initialized with all $\mathbf{0}$ elements, and the cluster indicator is initialized with a $K$-means clustering result of the input data samples.

**Optimization for $Y$** With $\Psi$ fixed, Eqn. (11) reduces to a trace maximization problem:

$$\max_{Y} \quad J = \text{trace}(Y^T X^T X Y) \tag{12}$$

Since $Y$ is an orthonormal matrix: $Y^T Y = I_K$, the spectral relaxation technique (Zha et al., 2001) can be adopted to compute the optimal $Y$. The solution is based on *Ky Fan matrix inequalities* below:

**Theorem.** *(Ky Fan) If $A$ be a symmetric matrix with eigenvalues $\{\lambda_1 \geq \lambda_2 \geq ... \geq \lambda_n\}$. Let the corresponding eigenvectors be $\{v_1, v_2, ...v_n\}$, then*

$$\max_{Y^T Y = I_K} \text{trace}(Y^T A Y) = \sum_{i=1}^{K} \lambda_i$$

*where the optimal $Y^*$ is given by $Y^* = [v_1, v_2, ...v_K]Q$ for any arbitrary orthogonal matrix $Q$.*

This spectral relaxation solution can be regarded as a manifold learning method that projects data samples from the original $d$-dimensional space to a new $K$-dimensional space. In our case, the $A$ matrix in *Ky Fan Theorem* takes the form of $X^T X$. In implementation, we first compute the $K$ largest eigenvectors of $X^T X$, and then apply the traditional $K$-means method, under the induced $K$-dimensional space, to compute the cluster assignments.

**Optimization for $\Psi$** With the $Y$ generated from Eqn. (12), Eqn. (11) becomes a trace minimization problem w.r.t. $\Psi$:

$$\min_{\Psi} \quad J = \text{trace}((X + \Psi G)^T (X + \Psi G)) - \text{trace}(Y^T (X + \Psi G)^T (X + \Psi G)Y) + \lambda \text{trace}(\Psi^T \Psi) \tag{13}$$

Through a careful investigation of the gradient and Hessian matrix of Eqn. (13), we found the $J$ could be proved to a smooth convex function, with its Hessian w.r.t. $\Psi$ being positive definite (PD) everywhere. Therefore, the only stationary point of $J$, where the gradient is evaluated to $\mathbf{0}$, locates the global minimum, and provides the optimal $\Psi^*$. The convexity proof is given as follows.

**Convexity proof of $J$ w.r.t. $\Psi$:** Firstly, we update $J$ in Eqn. (13), through several straightforward derivation steps (the details are given in Appendix A), to an equivalent form:

$$J = \text{trace}(X^T X) + 2\text{trace}(G^T \Psi^T X) + \text{trace}(\Psi G G^T \Psi^T) - \text{trace}(Y^T X^T XY)$$
$$- 2\text{trace}(Y^T G^T \Psi^T XY) - \text{trace}(\Psi G Y Y^T G^T \Psi^T) + \lambda\text{trace}(\Psi\Psi^T) \tag{14}$$

The gradient of $J$ w.r.t. $\Psi$ can then be computed as:

$$\frac{\partial J}{\partial \Psi} = 2XG^T - 2XYY^T G^T + 2\Psi G G^T - 2\Psi G Y Y^T G^T + 2\lambda\Psi \tag{15}$$

To facilitate the convexity proof, we rewrite this gradient equation as:

$$\frac{\partial J}{\partial \Psi} = N + \Psi M; \quad \text{where } N = 2XG^T - 2XYY^T G^T; \; M = 2(G(I - YY^T)G^T + \lambda I) \tag{16}$$

$N$ is a matrix of size $d \times n$. $M$ is a symmetric matrix of size $n \times n$, which can be proved positive definite, based on the theorem in (Horn & Johnson, 2012):

**Theorem.** *"Suppose that $A \in M_{m,n}$ and $B \in M_{n,m}$ with $m \leq n$. Then $BA$ has the same eigenvalues as $AB$, counting multiplicity, together with an additional $n - m$ eigenvalues equal to $0$."*

We know $Y^T * Y = I_K$, whose eigenvalues are all 1s. Then, according to this Theorem, the eigenvalues of $YY^T$ are 1s (multiplicity is $K$), and 0 (multiplicity is $n - K$). In the matrix $M$ of Eqn. (16), $I - YY^T$ is a positive semidefinite matrix as it is symmetric and its eigenvalues are either $0$ or $1$. $G$ is also positive definite because it is a kernel (Gram) matrix with the Gaussian kernel. With $G$ being symmetric PD and $\lambda$ setting to be a positive number in our algorithm, the matrix $M$ is guaranteed to be a PD matrix.

Expanding the gradient formulated in Eqn. (16) to individual elements of $\Psi$, it can be further written as:

$$\frac{\partial J}{\partial \Psi_{ij}} = N_{ij} + \sum_{u=1}^{n} \Psi_{iu} M_{uj} \qquad i \in d; j \in n \tag{17}$$

With Eqn. (17), Eqn. (16) can be resized into a vector of size $d \times n$. Then, the Hessian matrix of $J$ w.r.t. $\Psi$ can be calculated as below:

$$H = \begin{bmatrix}
M_{11}, M_{12}, ..., M_{1n}, 0, ................................................, 0 \\
M_{21}, M_{22}, ..., M_{2n}, 0, ................................................, 0 \\
. \\
. \\
M_{n1}, M_{n2}, ..., M_{nn}, 0, ................................................, 0 \\
0, ...................., 0, M_{11}, M_{12}, ..., M_{1n}, 0, ...................., 0 \\
0, ...................., 0, M_{21}, M_{22}, ..., M_{2n}, 0, ...................., 0 \\
. \\
. \\
0, ...................., 0, M_{n1}, M_{n2}, ..., M_{nn}, 0, ...................., 0 \\
\cdots \\
\cdots \\
0, ................................................, 0, M_{11}, M_{12}, ..., M_{1n} \\
0, ................................................, 0, M_{21}, M_{22}, ..., M_{2n} \\
. \\
. \\
0, ................................................, 0, M_{n1}, M_{n2}, ..., M_{nn}
\end{bmatrix}$$

It is clear that $H$ is a symmetric matrix with size $(d*n) \times (d*n)$. The diagonal of H is composed by $d$ repeating $M$ matrices. Let $\vec{z}$ be any non-zero column vector with size $(d*n) \times 1$. To prove $H$ is a

PD matrix, we want to show that $\vec{z}^T H \vec{z}$ is always positive. To this end, we rewrite $\vec{z}$ as $[\vec{z}_1, \vec{z}_2, ..., \vec{z}_d]$, where $\vec{z}_i$ is the sub-column of $\vec{z}$ with size $n \times 1$. Then $\vec{z}^T H \vec{z}$ can be computed as:

$$\vec{z}^T H \vec{z} = [\vec{z}_1^T M, \vec{z}_2^T M, ..., \vec{z}_d^T M] \vec{z} = \vec{z}_1^T M \vec{z}_1 + \vec{z}_2^T M \vec{z}_2 + ... + \vec{z}_d^T M \vec{z}_d \tag{18}$$

As $M$ has been proved to be a PD matrix, each item in Eqn. (18) is positive. Therefore, the summation $\vec{z}^T H \vec{z}$ is also positive. Since $\vec{z}$ is an arbitrary non-zero column vector, this shows $H$ is PD. With the Hessian matrix $H$ being PD everywhere, the objective function $J$ is convex w.r.t. $\Psi$. As a result, the stationary point of $J$ makes the unique global minimum solution $\Psi^*$. Let Eqn. (15) equal to 0, we get

$$\Psi^*(G(I - YY^T)G^T + \lambda I) = XYY^T G^T - XG^T \tag{19}$$

The matrix $M$ on the left is proved PD, thus invertible. The optimal solution of $\Psi$ is given as:

$$\Psi^* = (XYY^T G^T - XG^T)(G(I - YY^T)G^T + \lambda I)^{-1} \tag{20}$$

### 2.3 MAIN ALGORITHM

Based on the description above, our proposed CPD-UML algorithm can be summarized as the pseudo-code below:

---
**Algorithm 1** Main Algorithm of CPD-UML
---
**Input:** Samples $X$, cluster number $K$, regularization parameter $\lambda$, smoothness parameter $\sigma$ in CPD model, and threshold $\tau$

**Output:** Transformation matrix $\Psi$ and cluster indicator matrix $Y$
......................................................................................................
Initialize transformation matrix $\Psi$ using zero values;
Compute the initial cluster indicator matrix $Y$ using spectral relaxation;
**while** trace value changes $> \tau$ **do**
   Update $Y$ as in Section 2.2 (Ky Fan Theorem);
   Update $\Psi$ as in Eqn. (20);
   Compute the trace value in Eqn. (11);
**end (while)**
Return $\Psi$ and $Y$;

---

### 2.4 KERNELIZATION OF CPD-UML

So far, we developed and applied our proposed CPD-UML under input feature spaces. However, it can be further kernelized to improve the clustering performance for more complicated data. A kernel principal component analysis (KPCA) based framework (Zhang et al., 2010) is utilized in our work. After the input data instances are projected into kernel spaces introduced by KPCA, CPD-UML can be applied under the kernel spaces to learn both deformation field and clustering result, in the same manner as it is conducted under the original input spaces.

## 3 EXPERIMENTAL RESULTS

We performed experiments on a synthetic dataset and six benchmark datasets. Comparisons are made with state-of-the-art unsupervised metric learning solutions.

### 3.1 EXPERIMENTS ON SYNTHETIC DATASET

The two-moon synthetic dataset [1] was tested in the first set of experiments. It consists of two classes with 100 examples in each class. (see Fig. 1). All the samples were treated as unlabeled samples in the experiments. Both linear and kernel versions of our CPD-UML were tested.

---
[1]http://manifold.cs.uchicago.edu/manifold_regularization/data.html

**Linear version CPD-UML** In this experiment, our CPD-UML was applied in deforming the data samples to achieve better separability under the input space. The effectiveness of our approach is demonstrated by comparing with the base algorithm $K$-means.

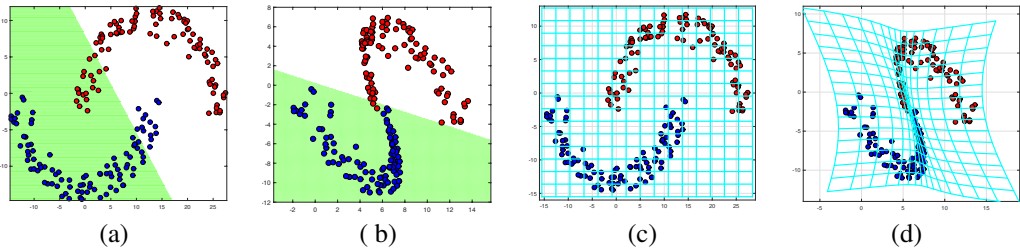

(a)  ( b)  (c)  (d)

Figure 1: (a) clustering result of K-means; (b) clustering result of CPD-UML; (c) and (d) show the deformation field of (b). Color figures are best viewed on screen.

The clustering results of $K$-means and CPD-UML are shown in Fig. 1 (a) and 1 (b) respectively. The sample labels are distinguished using blue and red colors. The clustering results are shown using the decision boundary. It is obvious that $K$-means cannot cluster the two-moon data well due to the data's non-separability under the input space. Our CPD-UML, on the contrary, achieves a 99% clustering accuracy by making the data samples linearly separable via space transformations. The deformation field of Fig. 1 (b) in the input space is shown in Fig. 1 (c) and (d). It is evident that our nonlinear metric learning model can deform feature spaces in a sophisticated yet smooth way to improve the data separability.

**Kernel version CPD-UML** In this set of experiments, various RBF kernels were applied on the two-moon dataset to simulate linearly non-separable cases under kernel spaces. The clustering results of kernel $K$-means with different RBF kernels (width = 4, 8, 16, 32) are shown in Fig. 2 (a) – 2 (d). Colors and decision boundaries stand for the same meaning as those in Fig. 1. Obviously, the performance of kernel $K$-means was getting worse with sub-optimal kernels, as in 2 (b), 2 (c) and 2 (d). Searching for an optimal RBF kernel requires cross-validation among many candidates, which could result in a large number of iterations. This procedure can be greatly eased by our kernel CPD-UML. The CPD transformation under kernel spaces provides a supplementary force to the kernelization to further improve the data separability, the same as it performs under the input space. Fig. 2 (f) – 2 (h) demonstrate the effectiveness of our CPD-UML. Same RBF kernels as in Fig. 2 (b) – 2 (d) were used, but better clustering results were obtained. The ability to work with sub-optimal kernels should also be regarded as a computational advantage of our model.

## 3.2 EXPERIMENTS ON BENCHMARK DATASETS

**Experimental Setup** In this section, we employ six benchmark datasets to evaluate the performance of our CPD-UML. They are five UCI datasets [2] : Breast, Diabetes, Cars, Dermatology, E. Coli and the USPS_20 handwritten data. Their basic information is summarized in Appendix B.

Both linear and kernel versions of our proposed approach were tested. For linear version, $K$-means method was used as the baseline for comparison. In addition, three unsupervised metric learning solutions, AML (Ye et al., 2007), RPCA-OM (Nie et al., 2014) and FME (Nie et al., 2010) were utilized as the competing solutions. For kernel version, the baseline algorithm is kernel $K$-means. NAML (Chen et al., 2007), the kernel version of AML is adopted. Since RPCA-OM and FME do not have their kernel version, the same kernelization strategy in 2.4 was applied to kernelize these two solutions. RBF kernels were applied for all kernel solutions.

Each dataset was partitioned into seen and unseen data randomly. Optimal cluster centers and parameters are determined by the seen data. Clustering performance is evaluated via the unseen data, which are labeled directly based on their distances away from the cluster centers. Similar setups have been used in (Nie et al., 2011; Huang et al., 2015). In the experiments, we performed 3-fold cross validation, in which two folds were used as seen data and one fold as unseen data. In the competing

---

[2]http://archive.ics.uci.edu/ml/

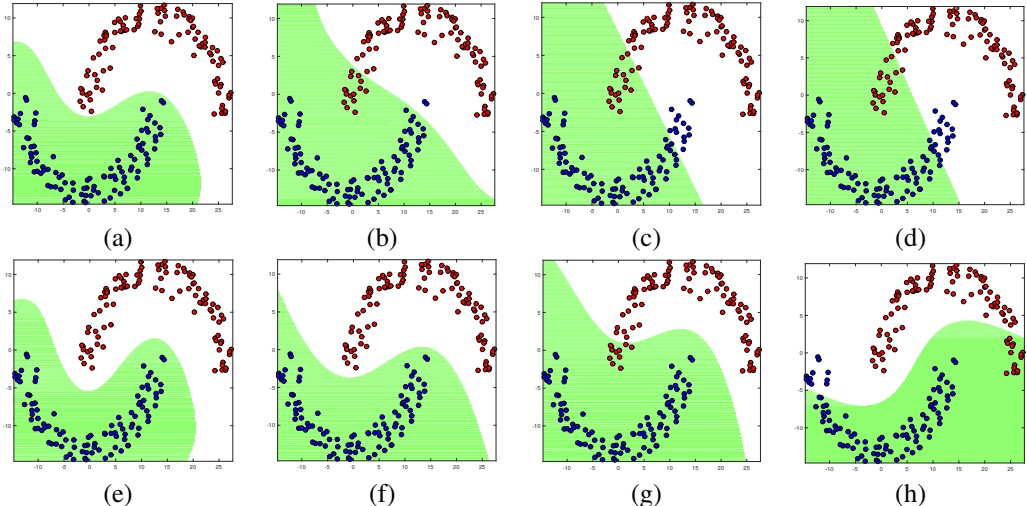

Figure 2: First row: clustering results of kernel K-means with RBF kernels width = 4, 8, 16 and 32. Second row: results of kernel version CPD-UML with RBF kernels width = 4, 8, 16 and 32.

solutions, the hyper-parameters were searched within the same range as in their publications. In our proposed approach, the regularization parameter $\lambda$ and smooth parameter $\sigma$ were searched from $\{10^0 \sim 10^{10}\}$ and $\{2^0 \sim 2^{10}\}$, respectively. The RBF kernel width for all kernel methods is chosen from $\{2^{-5} \sim 2^{10}\}$. Since the performance of tested methods depends on the initialization clusters, the clustering result of $K$-means was applied as the initialization clusters for all the competing solutions in each run. The performance of each algorithm was calculated over 20 runs.

**Results** We measured the performance using the ground truth provided in all six benchmark datasets. Three standard performance metrics were calculated: accuracy, normalized mutual information and purity. To better compare the tested methods in statistic, we conducted a Student's $t$-test with a $p$-value 0.05 between each pair of solutions for each dataset. The solutions were ranked using a scoring schema from (Wang et al., 2012). Compared with other methods, an algorithm scores 1 if it performs significantly better than one opponent in statistic; 0.5 if there is no significant difference, and 0 if it is worse.

Tables 1, 2 and 3 summarize the clustering performance and ranking scores. The best performance is identified in Boldface for each dataset. It is evident that our CPD-UML outperforms other competing solutions in all three standard measurements with significant margins. Highest ranking scores in the performance tables are all achieved by our kernel version approach. In addition, significant improvements have been obtained by our proposed approach compared with the baseline algorithm $K$-means and kernel $K$-means. It is also noteworthy that, the linear CPD-UML achieved comparable results with the other competing methods using RBF kernels, which further demonstrates the effectiveness of our nonlinear feature space transformation.

## 4 CONCLUSIONS

The proposed CPD-UML model learns a nonlinear metric and the clusters for the given data simultaneously. The nonlinear metric is achieved by a globally smooth nonlinear transformation, which improves the separability of given data during clustering. CPD is used as the transformation model because of its capability in deforming feature space in sophisticated yet smooth manner. Evaluations on synthetic and benchmark datasets demonstrate the effectiveness of our approach. Applying the proposed approach to other computer vision and machine learning problems are in the direction of our future research.

Table 1: Accuracy

| Algorithms | Breast | Diabetes | Cars | Dermatology | E. Coli | USPS20 | Total Score |
|---|---|---|---|---|---|---|---|
| $K$-means | $96.39 \pm 0.34$ (2.5) | $96.42 \pm 0.12$ (3.5) | $42.67 \pm 0.35$ (0.0) | $69.64 \pm 7.73$ (4.5) | $57.11 \pm 3.23$ (6.0) | $62.29 \pm 3.12$ (3.0) | 19.5 |
| AML | $96.34 \pm 0.35$ (1.5) | $96.49 \pm 0.09$ (4.5) | $43.67 \pm 0.50$ (1.5) | $63.17 \pm 3.07$ (0.0) | $58.87 \pm 3.27$ (6.0) | $64.36 \pm 3.93$ (4.0) | 17.5 |
| RPCA-OM | $96.54 \pm 0.49$ (3.5) | $96.71 \pm 0.31$ (6.0) | $43.81 \pm 0.78$ (1.5) | $71.15 \pm 4.50$ (5.0) | $58.55 \pm 3.28$ (6.0) | $62.64 \pm 3.87$ (3.5) | 25.5 |
| FME | $96.36 \pm 0.30$ (1.5) | $96.57 \pm 0.35$ (4.5) | $48.86 \pm 2.11$ (4.5) | $77.72 \pm 3.68$ (8.0) | $58.59 \pm 3.54$ (6.0) | $66.27 \pm 4.09$ (6.0) | 30.5 |
| CPD-UML | $\mathbf{97.06 \pm 0.44}$ (8.5) | $97.52 \pm 0.19$ (8.0) | $45.33 \pm 0.73$ (3.0) | $68.67 \pm 7.48$ (4.0) | $57.01 \pm 3.67$ (5.5) | $64.78 \pm 3.64$ (4.5) | 33.5 |
| Kernel $K$-means | $96.56 \pm 0.29$ (4.0) | $95.57 \pm 0.13$ (1.0) | $61.71 \pm 0.08$ (6.0) | $64.98 \pm 2.02$ (1.5) | $55.04 \pm 2.49$ (3.0) | $56.14 \pm 4.98$ (0.5) | 16.0 |
| NAML | $96.56 \pm 0.30$ (4.0) | $95.57 \pm 0.15$ (1.0) | $62.13 \pm 0.54$ (7.0) | $66.60 \pm 2.01$ (3.5) | $49.62 \pm 2.82$ (0.5) | $57.76 \pm 3.82$ (0.5) | 16.5 |
| $r$-RPCA-OM | $96.76 \pm 0.27$ (6.5) | $95.57 \pm 0.14$ (1.0) | $49.55 \pm 2.53$ (4.5) | $65.69 \pm 3.75$ (2.5) | $52.72 \pm 5.26$ (2.0) | $67.44 \pm 4.29$ (7.5) | 24.0 |
| $r$-FME | $96.73 \pm 0.25$ (5.0) | $96.79 \pm 0.14$ (6.5) | $63.38 \pm 1.51$ (8.5) | $79.45 \pm 3.86$ (8.0) | $50.78 \pm 2.53$ (1.0) | $68.13 \pm 5.47$ (7.5) | 36.5 |
| $r$-CPD-UML | $96.95 \pm 0.41$ (8.0) | $\mathbf{97.75 \pm 0.10}$ (9.0) | $\mathbf{63.77 \pm 1.52}$ (8.5) | $\mathbf{80.37 \pm 4.80}$ (8.0) | $\mathbf{63.02 \pm 2.74}$ (9.0) | $\mathbf{68.58 \pm 2.49}$ (8.0) | **50.5** |

Table 2: Normalized Mutual Information

| Algorithms | Breast | Diabetes | Cars | Dermatology | E. Coli | USPS20 | Total Score |
|---|---|---|---|---|---|---|---|
| $K$-means | $76.10 \pm 1.73$ (2.5) | $76.31 \pm 0.63$ (1.0) | $18.17 \pm 0.50$ (5.5) | $80.91 \pm 4.96$ (4.0) | $56.77 \pm 2.14$ (7.5) | $61.73 \pm 1.49$ (3.0) | 23.5 |
| AML | $75.83 \pm 1.80$ (1.5) | $76.58 \pm 0.11$ (1.5) | $\mathbf{20.53 \pm 0.77}$ (8.5) | $79.15 \pm 2.80$ (3.0) | $57.81 \pm 2.07$ (7.5) | $62.51 \pm 3.25$ (3.5) | 25.5 |
| RPCA-OM | $77.07 \pm 2.52$ (3.5) | $77.88 \pm 1.64$ (5.5) | $20.04 \pm 0.64$ (7.5) | $82.27 \pm 4.29$ (6.0) | $51.22 \pm 2.42$ (4.0) | $60.78 \pm 2.40$ (2.5) | 29.0 |
| FME | $75.93 \pm 1.50$ (1.5) | $77.01 \pm 1.77$ (3.0) | $19.45 \pm 2.44$ (8.0) | $84.75 \pm 3.58$ (8.0) | $56.73 \pm 2.76$ (7.5) | $64.01 \pm 2.73$ (6.0) | 34.0 |
| CPD-UML | $\mathbf{79.63 \pm 2.35}$ (8.5) | $83.31 \pm 0.64$ (8.0) | $18.19 \pm 0.51$ (5.5) | $79.46 \pm 5.09$ (4.0) | $46.98 \pm 8.29$ (2.0) | $61.12 \pm 3.32$ (2.5) | 30.5 |
| Kernel $K$-means | $77.01 \pm 1.58$ (4.0) | $77.06 \pm 0.10$ (3.5) | $8.50 \pm 0.05$ (3.0) | $74.65 \pm 3.62$ (0.0) | $51.04 \pm 1.76$ (4.0) | $59.48 \pm 3.05$ (1.5) | 16.0 |
| NAML | $77.01 \pm 1.58$ (4.0) | $77.06 \pm 0.11$ (3.5) | $1.50 \pm 0.95$ (0.0) | $79.86 \pm 1.43$ (3.0) | $45.66 \pm 4.03$ (1.5) | $60.87 \pm 3.11$ (2.5) | 14.5 |
| $r$-RPCA-OM | $78.19 \pm 1.50$ (6.5) | $77.08 \pm 1.47$ (3.5) | $3.30 \pm 1.90$ (1.0) | $79.84 \pm 2.66$ (3.5) | $18.36 \pm 1.85$ (0.0) | $66.12 \pm 2.66$ (8.0) | 22.5 |
| $r$-FME | $77.95 \pm 1.37$ (5.0) | $78.32 \pm 0.79$ (6.5) | $6.38 \pm 0.62$ (2.0) | $\mathbf{85.96 \pm 3.17}$ (8.5) | $49.74 \pm 2.89$ (3.5) | $67.02 \pm 3.76$ (8.0) | 33.5 |
| $r$-CPD-UML | $79.25 \pm 2.43$ (8.0) | $\mathbf{85.76 \pm 0.81}$ (9.0) | $10.12 \pm 2.26$ (4.0) | $81.46 \pm 2.65$ (5.0) | $\mathbf{57.85 \pm 2.18}$ (7.5) | $65.34 \pm 2.01$ (7.5) | **41.0** |

Table 3: Purity

| Algorithms | Breast | Diabetes | Cars | Dermatology | E. Coli | USPS20 | Total Score |
|---|---|---|---|---|---|---|---|
| $K$-means | $96.39 \pm 0.34$ (2.5) | $96.42 \pm 0.12$ (3.5) | $62.85 \pm 0.36$ (4.5) | $81.36 \pm 4.45$ (4.0) | $81.17 \pm 1.58$ (7.5) | $69.78 \pm 2.54$ (3.5) | 25.5 |
| AML | $96.34 \pm 0.35$ (1.5) | $96.49 \pm 0.09$ (4.5) | $62.91 \pm 0.38$ (4.5) | $78.94 \pm 1.81$ (2.5) | $81.39 \pm 1.42$ (7.5) | $71.08 \pm 4.32$ (4.5) | 25.0 |
| RPCA-OM | $96.54 \pm 0.49$ (3.5) | $96.71 \pm 0.31$ (6.0) | $63.85 \pm 0.80$ (7.0) | $82.64 \pm 3.85$ (5.0) | $74.50 \pm 1.85$ (2.5) | $69.60 \pm 3.58$ (3.5) | 27.5 |
| FME | $96.36 \pm 0.30$ (1.5) | $96.57 \pm 0.35$ (4.5) | $63.86 \pm 1.07$ (7.0) | $85.27 \pm 3.20$ (8.0) | $80.43 \pm 2.11$ (7.0) | $72.68 \pm 3.37$ (7.0) | 35.0 |
| CPD-UML | $\mathbf{97.06 \pm 0.44}$ (8.5) | $97.52 \pm 0.19$ (8.0) | $63.10 \pm 0.49$ (4.5) | $81.02 \pm 4.45$ (3.5) | $74.88 \pm 5.82$ (3.5) | $69.80 \pm 3.61$ (3.5) | 31.5 |
| Kernel $K$-means | $96.56 \pm 0.29$ (4.0) | $95.57 \pm 0.13$ (1.0) | $62.16 \pm 0.05$ (1.0) | $76.80 \pm 1.94$ (0.0) | $76.03 \pm 1.57$ (4.0) | $65.29 \pm 4.29$ (0.5) | 10.5 |
| NAML | $96.56 \pm 0.30$ (4.0) | $95.57 \pm 0.15$ (1.0) | $62.21 \pm 0.32$ (1.0) | $79.37 \pm 1.02$ (3.0) | $70.90 \pm 3.93$ (1.0) | $66.69 \pm 3.38$ (0.5) | 10.5 |
| $r$-RPCA-OM | $96.76 \pm 0.27$ (6.5) | $95.57 \pm 0.14$ (1.0) | $62.29 \pm 0.41$ (1.0) | $79.26 \pm 2.16$ (3.0) | $52.84 \pm 5.29$ (0.0) | $73.46 \pm 2.95$ (7.5) | 19.0 |
| $r$-FME | $96.73 \pm 0.25$ (5.0) | $96.79 \pm 0.14$ (6.5) | $63.38 \pm 1.76$ (5.5) | $85.44 \pm 3.16$ (8.0) | $76.01 \pm 2.62$ (4.0) | $72.77 \pm 3.42$ (7.0) | 36.0 |
| $r$-CPD-UML | $96.95 \pm 0.41$ (8.0) | $\mathbf{97.75 \pm 0.10}$ (9.0) | $\mathbf{64.86 \pm 0.85}$ (9.0) | $\mathbf{85.59 \pm 2.56}$ (8.0) | $\mathbf{82.09 \pm 1.74}$ (8.0) | $\mathbf{74.44 \pm 2.14}$ (7.5) | **49.5** |

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

## A APPENDIX A

In section 2.2, we optimize the object function $J$ in Eqn. (13) by computing its gradient and Hessian matrix w.r.t. $\Psi$. The Hessian matrix can be proved to be positive definite. The derivations from Eqn. (13) to Eqn. (14) is presented in this appendix. Eqn. (13) starts as:

$$\min_{\Psi} \quad J = \text{trace}((X + \Psi G)^T (X + \Psi G)) - \text{trace}(Y^T (X + \Psi G)^T (X + \Psi G)Y) + \lambda \text{trace}(\Psi^T \Psi)$$

It can be expanded to:

$$
\begin{aligned}
J = \text{trace}(&X^T X + X^T \Psi G + G^T \Psi^T X + G^T \Psi^T \Psi G) \\
- \text{trace}(&Y^T X^T XY + Y^T X^T \Psi GY + Y^T G^T \Psi^T XY \\
+ &Y^T G^T \Psi^T \Psi GY) + \lambda \text{trace}(\Psi^T \Psi)
\end{aligned}
\tag{21}
$$

As $\text{trace}(A) = \text{trace}(A^T)$ for any matrix $A$, we get $\text{trace}(X^T \Psi G) = \text{trace}(G^T \Psi^T X)$ and $\text{trace}(Y^T X^T \Psi GY) = \text{trace}(Y^T G^T \Psi^T XY)$. With these two equations, Eqn. (21) becomes:

$$
\begin{aligned}
J = \text{trace}(&X^T X) + 2\text{trace}(G^T \Psi^T X) + \text{trace}(G^T \Psi^T \Psi G)) \\
- \text{trace}(&Y^T X^T XY) - 2\text{trace}(Y^T G^T \Psi^T XY) \\
- \text{trace}(&Y^T G^T \Psi^T \Psi GY) + \lambda \text{trace}(\Psi^T \Psi)
\end{aligned}
\tag{22}
$$

Through some simple matrix manipulations, as well as based on the fact that $\text{trace}(AB) = \text{trace}(BA)$ for any matrix $A$ and $B$, Eqn. (22) can be updated to Eqn. (14):

$$
\begin{aligned}
J = \text{trace}(&X^T X) + 2\text{trace}(G^T \Psi^T X) + \text{trace}(\Psi GG^T \Psi^T) - \text{trace}(Y^T X^T XY) \\
- &2\text{trace}(Y^T G^T \Psi^T XY) - \text{trace}(\Psi GYY^T G^T \Psi^T) + \lambda \text{trace}(\Psi \Psi^T)
\end{aligned}
$$

## B APPENDIX B

In section 3.2, we employ six benchmark datasets to evaluate the performance of our CPD-UML. They are five UCI datasets: Breast, Diabetes, Cars, Dermatology, E. coli and the USPS_20 handwritten data. Their basic information is summarized in Table 4.

Table 4: Six benchmark datasets used in experiments. Columns show the name, numbers of samples, attributes and classes of each dataset.

| Datasets | # Samples | # Attributes | # Classes |
|----------|-----------|--------------|-----------|
| Breast | 683 | 10 | 2 |
| Diabetes | 768 | 8 | 2 |
| Cars | 392 | 8 | 3 |
| Dermatology | 366 | 34 | 6 |
| Ecoli | 336 | 343 | 8 |
| USPS20 | 1854 | 256 | 10 |

