# OpenReview forum: "UNSUPERVISED METRIC LEARNING VIA NONLINEAR FEATURE SPACE TRANSFORMATIONS"
_ICLR.cc/2018/Conference — Reject_

### Official Review · AnonReviewer3 · 2017-11-21
**A well-written metric learning paper with a bit traditional idea**

**Rating:** 6
**Confidence:** 4

**Review:**

This paper proposed an unsupervised metric learning method, which is designed for clustering and cannot be used for other problems. The authors argued that unsupervised metric learning should not be a pre-processing method for the following clustering method due to the lack of any similarity/dissimilarity constraint. Consequently the proposed formulation plugs a certain metric learning objective, which is called CPD given in (1) and (3), into the k-means objective (2). After some linear algebra, it arrives the objective in (10) or its regularized version in (11). In order to solve (11), an alternative optimization is used to iteratively obtain the optimal Y given fixed Psi and obtain optimal Psi given fixed Y. More than one page of space is for proving the convexity of the latter subproblem.

The paper is overall well-written and I have only 1 question about the clarity: there is a very short Sec. 2.4 saying that "So far, we developed and applied our proposed CPD-UML under input feature spaces. However, it can be further kernelized to improve the clustering performance for more complicated data." I found this quite confusing. Just after (1), it is mentioned that Psi is the weight matrix for the Gaussian kernel functions and g is the Gaussian kernel function; it is also mentioned between (16) and (17) that G is a kernel (Gram) matrix with the Gaussian kernel. This issue of inconsistency should be clarified.

The main issues of the paper are the motivation and the experiments. While the argument of the authors is partially true, it is quite difficult to distinguish unsupervised metric learning from unsupervised feature/deep learning nowadays. The CPD model is limited in the nonlinearity to me: as shown in (1), the nonlinear function v is nonlinear in x via the so-called empirical kernel map G, and more importantly v is linear in its parameters namely Psi. If we would like to use a nonlinear-in-parameter model such as deep networks for v, the optimization for Y still works but the optimization for Psi can no longer work. This means the proposed learning objective is not model-free.

The experiments are correct to me, where 3 performance measures of clustering are reported for 10 methods on 6 datasets. However, all the datasets are quite small and thus cannot represent the most reliable comparisons of these methods. Moreover, the computational complexity of the proposed method is not discussed, but I guess it is quite high since both of the alternative subproblems require eigenvalue decomposition or solving a linear system.

---

### Official Review · AnonReviewer1 · 2017-11-27
**Interesting idea, but lacks technical clarity**

**Rating:** 4
**Confidence:** 5

**Review:**

This paper presents a scheme for unsupervised metric learning using coherent point drifting (CPD)-- the core idea is to learn a parametric model of CPD that shifts the input points such that the shifted points lead to better clustering in a K-Means setup. Following the work of Myronenko & Song, 2010, this paper uses a linear parametric model for the drift (in CPD) after mapping the input points to a kernel feature space using an RBF kernel. The CPD model is directly used within the KMeans objective -- the drift parameter matrix and the KMeans cluster assignment matrix are jointly learned using block-coordinate descent (BCD). The paper uses some interesting properties of the CPD model to derive an efficient optimization solver for the BCD subproblems. Experiments are provided on UCI datasets and demonstrate some promise.

Pros:
1) The idea of using CPD for unsupervised metric learning is quite interesting
2) The exploration into the convexity of the CPD parameter learning -- although straightforward -- is also perhaps interesting.
3) The experiments show some promise.

Cons:
1) Lacking motivation/Intuition
The main motivation for the approach, as far as I understand, is to learn cluster boundaries for non-linear data -- where K-Means fails. However, it is unclear to me why would one need to use K-Means for non-linear data, why not use kernelized kmeans? The proposed CPD model also is essentially learning a linear transformation of the kernelized feature space. So in contrast to kernelized kmeans, what is the advantage of the proposed framework? I see there is an improvement in performance compared to kernelized kmeans, however, intuitively I do not see how that improvement comes from? Perhaps providing some specific examples/scenarios or graphic illustrations will help appreciate the method.

2) Novelty/Significance
I think the novelty of this paper is perhaps marginal. The main idea is to directly use CPD from a prior work in a KMeans setup. There are a few parameters to be estimated in the joint learning objective, for which a block-coordinate descent strategy is proposed. The derivations are perhaps straightforward. As noted above, it is not clear what is the significance of this combination or how does it improve performance. As far as CPD goes, it looks to me that the performance depends heavily on the choice of the Gaussian RBF bandwidth parameter, and it is not clear to me how such a parameter can be selected in a unsupervised setting, when class labels are not available for cross-validation. The paper does not provide any intuitions on this front.

3) Technical details.
There are a few important details that I do not quite follow in the paper.

a) The CPD is originally designed for the point matching problem, and its parametric form (\Psi) is derived using a different a Tikhonov regularized regression model as described just above (1). The current paper directly uses this parametric form in a KMeans setup and solve the resultant problem jointly for the CPD parameter and the clustering assignment. However, it is not clear to me how the paper could use the optimal parametric form for Tikhonov regression as the optimum for the clustering problem. Ideally, I would think when formulating the joint optimization for the clustering problem, the optimal functional v(x) should also be learned/derived for the clustering problem, or some proof should be provided showing the functionals are the same. Without this, I am not convinced that the proposed formulation indeed learns the optimum drifts and the clusters jointly.

b)  The subproblem on Y (the assignment matrix) looks like a standard SVD objective. It is not clear why would it be necessary to resort to Ky Fan's theorem for its optimal solution.

c) The paper talks about manifold embedding in the abstract and in Sec. 2.2. However, it appears to be a straightforward dimensionality reduction (PCA) of data. If not, what is the precise manifold that is described here?

d) Eq. 9, the definition of Y_c is incorrect and unclear. p is defined as a vector of ones, earlier.

e) Although the assignment matrix Y has orthogonal columns, it is a binary matrix. If it is approximated by an orthonormal frame, how do you reduce it to a binary matrix? Does taking the largest values in each column suffice -- it does not look like so. However, in the paper, Y is relaxed to an orthonormal frame, which is estimated using PCA, the data points are then projected onto this low-dimensional subspace, and then k-means applied to get the Y matrix. The provided math does not support any of these steps. Thus, the technical exposition is imprecise and the solutions appear rather heuristic.

f) The kernelized variant of the proposed scheme, described in Sec. 2.4 is missing important details. How precisely is the kernelization done? How is CPD extended to that setup and what would be the Gaussian kernel G in that case, and what does \Psi signify?

g) Figure 2, it seems that kernel kmeans and the proposed CPD-UML show similar cluster boundaries for low-kernel widths. Why are the high kernel widths beneficial?

4) Experiments
There is some improvement of the proposed method -- however overall, the improvements are marginal. The discussion is missing any analysis of the results. Why it works at times, how well it improves on kernelized kmeans, and why? What is the advantage over other competitive schemes, etc.

In summary, while there is a minor novelty in connecting two separate ideas (CPD and UML) into a joint UML setup, the paper lacks sufficient motivations for proposing this setup (in contrast to say kernelized kmeans), the technical details are unconvincing, and the experiments lack sufficient details or analysis. Thus, I do not think this paper is ready to be accepted in its current form.

---

### Official Review · AnonReviewer2 · 2017-11-28
**The paper proposed a new**

**Rating:** 4
**Confidence:** 4

**Review:**

This paper proposed a nonlinear unsupervised metric learning framework. The authors combine Coherent Point Drifting and the k-means approaches under the trace minimization framework. However, I am afraid that the novelty and insight of this work is not good enough for acceptance.

Pros:
The paper is well written and easy to follow.

Cons:
1 The novelty of this paper is limited.
The authors mainly combine Coherent Point Drifting and the k-means under the trace minimization framework. The trace minimization is then solved with an EM-like iterative minimization.
However, trace minimization is already well explored and this paper provides little insight. Furthermore, there is not any theoretical guarantee how this iterative minimization approach will converge to.

2 For a method with limited novelty, comprehensive experiments are needed to verify its effectiveness. However, the experimental setting of this paper is biased.
An important line of works, namely deep learning based clustering, are totally missing.
Comprehensive experiments with other deep learning based clustering are required.

---

### Decision · Program_Chairs · 2018-01-29
**ICLR 2018 Conference Acceptance Decision**

**Decision:**

Reject

**Comment:**

The paper is well written overall. However, the algorithmic framework has limited novelty and the reviewers unanimously are unconvinced by experimental results showing marginal improvements on smallish UCI datasets.